# Numerical Simulation of Swirl Flow Characteristics of CO_2_ Hydrate Slurry by Short Twisted Band

**DOI:** 10.3390/e23070913

**Published:** 2021-07-18

**Authors:** Yongchao Rao, Zehui Liu, Shuli Wang, Lijun Li, Qi Sun

**Affiliations:** 1School of Petroleum Engineering, Changzhou University, Changzhou 213164, China; ryc@cczu.edu.cn (Y.R.); 15061960380@smail.cczu.edu.cn (Z.L.); 19082003350@smail.cczu.edu.cn (L.L.); 2Key Laboratory of Oil and Gas Storage and Transportation Technology of Jiangsu Province, Changzhou 213164, China; 3College of Chemistry and Chemical Engineering, Yulin University, Yulin 719000, China; sunqi@yulinu.edu.cn

**Keywords:** hydrate, swirl flow, flow characteristics, numerical simulation

## Abstract

The development of oil and gas resources is gradually transferring to the deep sea, and the hydrate plugging of submarine pipelines at high pressures and low temperatures is becoming an important problem to ensure the safety of pipeline operations. The swirl flow is a new method to expand the boundary of hydrate safe flow. Numerical simulation of the hydrate slurry flow characteristics in a horizontal pipeline by twisted band has been carried out, and the flow of CO_2_ hydrate slurry in low concentration has been simulated by the RSM and DPM models. The results show that the heat transfer efficiency is also related to Re and particle concentration. The velocity distribution has the form of symmetrical double peaks, and the peaks finally merge at the center of the pipeline. Vortexes firstly appear on both sides of the edge of the twisted band, and then move to the middle part of the twisted band. Finally, the vortex center almost coincides with the velocity center. The rotation direction of hydrate particles is the same as the twisted direction of the twisted band, twist rate (Y) is smaller, Re is larger, and the symmetric vortex lines merge farther away. The initial swirl number is mainly related to Y, but not Re. The swirl flow attenuates exponentially, and its attenuation rate is mainly related to Re, but not Y. Compared with ordinary pipelines, the swirl flow can obviously improve the transportation distance of hydrate slurry.

## 1. Introduction

Plugging problems in mixed natural gas hydrate pipelines are become more and more serious. Hydrate blockages block the partial pressure of the pipeline, causing damage to pipeline equipment. The control of hydrate plugging is thus gradually becoming an important problem to ensure the safety of pipeline operations. The traditional methods not only cost a lot, but also cause pollution to the environment. The current research is not focused on hydrate suppression, but on the generation of hydrate to ensure the safe flow of hydrate to achieve the goal of risk control. Swirl flow has strong carrying capacity, and it can enhance the heat transfer between fluids, which has important guiding significance for the safe transportation of gas hydrate.

The research on two-phase swirl flows is also very active at home and abroad. The flow characteristics of swirl flow are studied by experiment and numerical simulation. The structural parameters of the swirler have an important influence on the swirl flow. The heat transfer characteristics and pressure drop of turbulent flow have been measured in a propeller vortex generator and multi-pitch spring pipeline, and it was found that the position of the propeller and swirl spring pitch had an impact on heat transfer enhancement, and the pressure of the propeller was increased. Compared with the swirl twisted band, the swirl strength caused by the swirl guide plate was larger and the attenuation was slower. The swirl number was strongly dependent on Reynolds number in complete laminar flow state, but weakly dependent on Reynolds number in turbulent flow conditions [1,2,3,4,5]. The attenuation of swirl number was mainly affected by Reynolds number rather than inlet swirl strength [6,7,8]. Under the same conditions, the nussel number in pipelines with built-in twisted bands was 1.73~3.85 times larger than that of light pipelines [9,10,11].

In terms of the simulation of hydrate slurry flow, the particle size distribution in the horizontal pipe is normally regular. The initial particle size is larger, while the final particle size from the normal distribution deviates more [12,13,14,15]. Liang [16,17] et al. used the DPM model to conduct a numerical simulation of heat transfer and deposition characteristics of NGH in a swirl flow system. Jassim [18] et al. used CFD software to simulate the flow of slurry in a gas pipeline and proposed a new model for hydrate deposition mechanisms. Shi [19] et al. first simulated the solid–liquid two-phase isothermal flow of TBAB hydrate in a horizontal 90-degree bend and U-shaped pipe by using Euler’s two-fluid model combined with the interphase heat and mass transfer model. Boris [20,21] et al. used CFD combined with the PBM model to simulate the turbulent flow of slurry in the three phases of oil, water, and gas hydrate. The model can predict the details of the formation and agglomeration of hydrate phases and the interactions between solids and particles formed in the hydrate phases. Song [22] et al. carried out a numerical simulation of the mechanical properties and mechanical stability of the hydrate deposition on the pipe wall. Li [23] et al. used the three-dimensional Euler model combined with the particle flow mechanics theory to consider the heat and mass transfer in the hydration decomposition process and simulated the gas–liquid–solid three-phase flow in the hydrate slurry decomposition process. Sateesh [24] et al. established a three-dimensional Euler–Euler particle multiphase turbulent CFD model considering the impact of particle collision, drag, lift, and turbulence diffusion.

In conclusion, academics have carried out numerical simulations and experimental research on the combination of pipeline flow law of hydrate, but studies on the heat transfer, pressure drop, and deposition law of hydrate slurry in swirl flow systems are few and far between. At the same time, the heat transfer of the spiral flow system, the laws of hydrate formation, and the influence of particle movement are also important interference factors. Therefore, the DPM model and RSM model are used. The movement of solid particles is used to numerically simulate the complex spiral flow characteristics of CO_2_ hydrates in the spiral flow pipeline in a short twisted band. The deposition and heat transfer characteristics of CO_2_ hydrate particles in the spiral flow pipeline have been studied. The velocity distribution, turbulence intensity, temperature distribution, vortex line distribution, attenuation law of swirl number, and deposition law of hydrate particles have been investigated using the computational fluid dynamics (CFD) technology.

## 2. Numerical Simulation Method

### 2.1. Physical Model

#### 2.1.1. Geometric Model

The twisted band is used as the swirler and is fixed at the entrance of the pipeline. After the gas-liquid phase passes through the twisted band, the phase interface is reconstructed under centrifugal force. Three short twisted bands with different twist (Y (the rate of axial length to width of a twisted band)) are used to compare the effects of different flow velocities, twist rates, and pressure drops. The physical model of the twisted band is shown in Figure 1 and Figure 2.

The twisted band is adopted as the swirl device, and its physical model is shown in Figure 3. Its diameter (D) is 25 mm and its length (L) is 2500 mm. The pipe can be divided into front and rear sections. The front section contains a twisted band structure. The twisted band is set at the entrance, and L_1_ is 400 mm. L_2_ is 2100 mm. The cartesian coordinate system is adopted for calculations. The origin of coordinates is located at the center of the inlet surface of the pipeline. The Y-axis is the flow direction and flows from the left end of the pipeline to the right end.

#### 2.1.2. Boundary Conditions

The density [25] of CO_2_ hydrate particles is 1116 kg/m^3^, the particle size is 0.001 mm, and particle volume concentration is 1%~8%. The medium is water with a density of 1000 kg/m^3^. The inlet boundary is the velocity inlet to flow horizontally along the pipeline. Export border is the outflow. The temperature of the hydrate is higher than the pipeline wall because of the heat release during the formation process. The inlet hydrate temperature is 280 K, and the wall and liquid temperature is 277 K. The pressure reference point is at the outlet center of the pipeline with a reference pressure of 0 Pa. The turbulence intensity is that the liquid phase flow velocity is substituted into the turbulence calculation formula, and the twisted band is fixed at the entrance of the pipeline, so the solid wall non-slip boundary is adopted. The effect of gravity on deposition is considered. It is assumed that hydrate particles are uniformly distributed at the pipeline entrance. The data parameters are shown in Table 1.

### 2.2. Meshing

The experimental model can be divided into two parts. The front part of the pipe is the swirl device, and the length of the twisted band is 400 mm. In order to meet the special requirements of the twisted band, the twisted band section is treated with unstructured mesh. The twisted band and the mesh at the edge of the pipe are encrypted. The thickness of the bottom layer is 0.2 mm, and 5 layers are encrypted according to a ratio of 1:1. The space between the other parts is divided by a body grid of 2 mm, and the result is shown in Figure 4. In order to improve the calculation accuracy and reduce the calculation time, an o-type structure grid is used to divide the back pipe segment. The mesh size is 2 mm and the wall surface is also encrypted. The grid division of the back pipe segment is shown in Figure 5.

### 2.3. Mathematical Model

#### 2.3.1. Governing Equations

The continuity equation is:(1)$$\frac{\partial \rho }{\partial t}+\frac{\partial }{\partial x_{i}}\left(\rho u_{i}\right)=0$$

The momentum equation is:(2)$$\frac{\partial }{\partial t}\left(\rho u_{i}\right)+\frac{\partial }{\partial x_{j}}\left(\rho u_{i}u_{j}\right)+\frac{\partial p}{\partial x_{i}}-\frac{\partial \tau_{ij}}{\partial x_{j}}-\frac{\partial \tau_{ij}^{-1}}{\partial x_{j}}=0$$
where, *ρ*, μ, and p are gas density, velocity, and static pressure, respectively; $\tau_{ij}$ is the viscous stress tensor; and t is the time.

The energy equation is:(3)$$\frac{\partial \left(\rho T\right)}{\partial t}+\frac{\partial \left(\rho uT\right)}{\partial x}+\frac{\partial \left(\rho vT\right)}{\partial y}+\frac{\partial \left(\rho wT\right)}{\partial z}=\frac{\partial }{\partial x}\left(\frac{\lambda }{c_{p}}\frac{\partial T}{\partial x}\right)+\frac{\partial }{\partial y}\left(\frac{\lambda }{c_{p}}\frac{\partial T}{\partial y}\right)+\frac{\partial }{\partial z}\left(\frac{\lambda }{c_{p}}\frac{\partial T}{\partial z}\right)$$

*ρ*, *c_p_*, T, and *λ* are liquid density, specific heat at constant pressure, temperature, thermal conductivity, respectively, and u, *v*, and w are speed.

#### 2.3.2. Discrete Phase Model

The discrete phase model (DPM) is selected to simulate the motion of hydrate particles in the twisted band because of their small particle size. It is generally believed that particles are mainly affected by gravity and drag forces. Compared to other forces, such as Saffman forces, gravity and drag forces larger by orders of magnitude. In order to improve the calculation efficiency, the influence of other forces is ignored. Due to the small particle and volume concentration, the collision between particles is ignored in the DPM model, and only the interaction between the liquid and particles is considered. The control equation of liquid and particle phase is solved alternately until the calculation converges. The motion equation of the discrete phase is expressed as follows:(4)$$\frac{du_{p}}{dt}=\frac{1}{\tau }\left(\bar{u}+u^{\prime }-u_{p}\right)$$
(5)$$\frac{d\upsilon_{p}}{dt}=\frac{1}{\tau }\left(\bar{\upsilon }+\upsilon^{\prime }-\upsilon_{p}\right)+\frac{w_{p}{}^{2}}{r_{p}}$$
(6)$$\frac{dw_{p}}{d^{2}t}=\frac{1}{\tau }\left(\bar{w}+w^{\prime }-w_{p}\right)+\frac{\upsilon_{p}w_{p}}{r_{p}}$$
where, $u_{p}$, $v_{p}$, and $w_{p}$ represent the axial, radial and tangential velocity of the particle, respectively; $\bar{u}$, $\bar{v}$, and $\bar{w}$ represent the time-averaged velocity in the axial, tangential, and radial direction, respectively; $u^{\prime }$, $v^{\prime }$, and $w^{\prime }$ represent the axial, tangential, and radial pulsation velocities, respectively; $r_{p}$ is the radial cylindrical coordinate of particle motion; and τ is the relaxation time of energy transfer in particle flow.
(7)τ=ρpdp218μ24CDRep
where, $\rho_{p}$ is particle density; $d_{p}$ is particle diameter; μ is the viscosity coefficient; $C_{D}$ is the drag force coefficient; and Rep is the relative Reynolds number of particles.

The particle orbits in the DPM model are calculated by the differential equation of particle force under the integral Laplace coordinate system. The equation of force equilibrium of particles (particle inertia = various forces acting on particles) in the cartesian coordinate system (x direction) is:(8)$$\frac{du_{p}}{dt}=F_{D}(u-u_{p})+\frac{g_{x}(\rho_{p}-\rho )}{\rho_{p}}+F_{X}$$
where $F_{X}$ is the additional acceleration term (force per unit particle mass) and $F_{D}(u-u_{p})$ is the drag force per unit particle mass. The formula is as follows:(9)$$F_{D}=\frac{18\mu }{\rho_{p}d_{p}^{2}}\frac{C_{D}Re}{24}$$

The particle Reynolds number (Re) can be expressed as follows:(10)Re=ρdp|up−u|μ

The drag force coefficient $C_{D}$ can be expressed as follows:(11)CD=24(1+b1Reb2)Re+b3Reb4+Re

Among them,
$$b_{1}=exp\left(2.3288-6.4581+2.4486\phi^{2}\right)$$
$$b_{2}=0.0964+0.5565\phi$$
$$b_{3}=exp\left(4.905-13.8944\phi +18.4222\phi^{2}-10.2599\phi^{3}\right)$$
(12)$$b_{4}=exp\left(1.4681+12.2584\phi -20.7322\phi^{2}+15.8855\phi^{3}\right)$$

ϕ is the shape coefficient, which is defined as follows:(13)$$\phi =\frac{s}{S}$$
where, *S* is the surface area of spherical particles with the same volume as the actual particles and S is the surface area of the actual particle. For spherical particles, ϕ=1.

### 2.4. Calculation Method

The software Fluent is used to simulate the unsteady two-phase flow of the pipeline, and the time step is 2 × 10^−4^ s. The turbulence model is the RSM model, and the DPM model is a two-phase flow model. The simulation is carried out at low pressure and flow velocity, so an implicit solver based on pressure is used. The finite volume method is used to discretize the equation. The SIMPLEC algorithm is used for pressure–velocity coupling. The convergence condition is that residual value <10^−6^. In addition, in order to improve the accuracy, the pressure equation, momentum equation, turbulent kinetic energy equation, and turbulent diffusivity equation are presented with the second-order upwind scheme.

### 2.5. Grid Independence Test

Mesh generation is an important step in the establishment of the finite element model. In this process, many problems need to be considered and the workload is very heavy. The unit length, quantity, and density of the grid have a direct influence on the precision and time of the calculation.

In order to meet the requirements of computational accuracy and ensure computational efficiency, 8 mm, 4 mm, and 2 mm mesh sizes are selected for grid independence verification. The simulation verification conditions: pipe diameter D = 24 mm, gas Re = 20,000, twist rate of twisted band = 6.2, and the initial concentration of particles = 2%. As shown in Figure 6, the velocity distribution on the cross-section at Z = 5D of the pipeline is selected for comparison. Under these three mesh size conditions, the velocity distribution curves obtained by the simulation are generally similar. However, compared with 2 mm (6,541,200 cells) and 4 mm (4,209,480 cells), the mesh size of 8 mm (2,865,700 cells) varies greatly, and there are fewer meshes near the wall, so the calculation accuracy is not good enough. On the other hand, grid sizes of 2 mm (6,541,200 cells) and 4 mm (4,209,480 cells) are basically the same, and the accuracy is not improved much near the wall. In order to improve the computing efficiency, a grid size of 4 mm (4,209 480 cells) is finally selected as the computational grid.

### 2.6. Experimental Verification

The simulation and experimental measurement of ice slurry are relatively perfect, and they have many similarities in many basic properties such as density. Therefore, Sari et al. [26]’s ice slurry flow experiment and the theoretical value derived by Kitanovski et al. [27] under similar working conditions are adopted as simulation verification. The experiment is carried out on a horizontal pipe with a diameter of 23 mm and a length of 1 m. The mass flow velocity is 0.5 kg/s, the particle size is 0.1 mm, and the fluid flow velocity is 1.5 m/s. Fluent is used to simulate the hydrate flow under the same working conditions. The simulation results of velocity field distribution at the same section are compared with the experimental results, as shown in Figure 7. The concentration distribution of hydrate slurry is compared with the theoretical value, as shown in Figure 8. The error between the simulation results and the experimental value and the theoretical prediction value is within 20%, so the model is reliable.

## 3. Results and Discussion

### 3.1. Velocity Distribution

The velocity magnitude and vector distribution at different sections of the pipeline are shown in Figure 9. From the figure, It can be seen that the peak velocity of the pipeline appears near the pipe wall at the beginning, showing a symmetrical bimodal structure. It can be seen from the velocity vector line that, with the increase in distance, the tangential velocity begins to change from swirl to ordinary two-phase flow, and it decreases continuously. The tangential velocity generated by the swirl flow is an important factor to ensure that the hydrate will not rapidly deposit and adhere to the pipe wall. After the twisted band, the two velocity centers finally merge rapidly, and the degree peak value gradually moves closer to the center of the pipe. The swirl flow changes to ordinary liquid-solid two-phase flow with a large number of hydrate deposits.

In order to better understand the changing process of velocity, the absolute velocity is broken down into axial velocity, radial velocity, and tangential velocity. The axial velocity distribution at each position at the center line of different sections is shown in Figure 10a. In the figure, it can be seen that the axial velocity has a bimodal structure with an “M” shape. Near the wall surface and at the edge of the twisted band, the velocity gradient is large due to the influence of the viscous bottom layer. Due to the distance from the swirler, the swirl flow continuously attenuates, and the velocity gradient in the rear segment starts to change from a bimonal-peak structure to a parabolic structure. As shown in Figure 8, the maximum axial velocity is only 1/3 of the incident velocity (the incident velocity is 3 m/s) when the twisted band has just started to spin. Then, the velocity increases rapidly, and the peak axial velocity is at 1/2r on the pipeline wall. The fluid is chaotic at the beginning due to the presence of the swirler, and the fluid velocity is redistributed. Then, the swirl flow stabilizes rapidly and the axial velocity increases. The radial velocity distribution at each position at the center line of different sections is shown in Figure 10b. The radial velocity peak position is close to the axial velocity peak position. The distance is farther, and the peak value is smaller, which indicates that the swirl flow becomes more and more stable. The radial velocity is caused by the fluid turbulence in the section of the twisted band. However, in general, the absolute value of radial velocity is relatively small, and the velocity rapidly decays to about 0 m/s after leaving the twisted band section. The tangential velocity distribution at each position at the center line of different sections is shown in Figure 10c. The tangential velocity in the twisted band section and the rear pipe section are of an “M” shape, which indicates that the swirl flow at the 24D position of the pipe has not been completely attenuated. The peak value of tangential velocity is 1/6~1/7r away from the pipeline wall, and the peak value of tangential velocity is about 1/3~1/4 of the incident velocity. Tangential velocity is large, and it is the tangential velocity that generates the tangential force between particles near the pipe and between particles and the pipe wall. Meanwhile, the particles themselves swirl so that they are not easy to agglomerate and deposit. Tangential velocity “levitates” particles, while axial velocity “migrates” particles. The combination of the two expands the transportation distance of hydrate particles.

### 3.2. Turbulence Intensity

The turbulence intensity reflects the rate of the pulsation velocity at a certain point to the average velocity. The turbulence intensity is greater when the doping is more uniform. The calculation formula of turbulence intensity is as follows:(14)$$I=0.16{\left(Re\right)}^{-\frac{1}{8}}$$
where *I* is the turbulence intensity and Re is the Reynolds number.

The distribution curve of turbulence intensity in the starting part of the twisted band is shown in Figure 11a. It can be seen from the figure that the turbulence intensity distribution curve is roughly a “W” type inverted bimodal structure at the horizontal diameter of the pipe. The turbulence intensity near the pipeline wall and the twisted band is higher, and the turbulence intensity is lower in the middle. The combined vortex structure of the swirl flow is composed of the core area and the annular area. The rigid main vortex, located in the core area, is stable and the turbulence intensity is small. The free vortex at the edge wall is small but unstable, so the fluid disturbance in the annular region is larger and the turbulence intensity is greater than that in the core region. The vortex located in the core area is less stable when it starts to form, so the turbulence intensity first increases, which reaches a maximum of 8%, and then decreases after stabilization. The core also slowly moves closer to the wall. The distribution curve of turbulence intensity in the section outside the twisted band is shown in Figure 11b. After detachment from the twisted band, the turbulence intensity at the center of the pipe decreases rapidly, and the turbulence intensity distribution curve begins to change from a “W” to a “U”. The turbulence intensity at the center decreases at firstly as the free vortex area near the wall of the twisted band begins to transform to the core vortex area. Then the turbulence intensity in the central zone increases again, and the two core vortices start to merge into one core vortex, and the velocity direction at the intersection is the opposite, which leads to an increase in the pulsation velocity.

### 3.3. Temperature Distribution

The temperature distribution of hydrate particles at different sections of the pipeline under the conditions of Re = 9000 and Y = 8.8 is shown in Figure 12. It can be seen from the figure that there are two symmetric high temperature regions at first. With the increasing distance from the swirler, the two high temperature regions gradually close to form a “concentric circle”. The intermediate temperature is high and gradually decreases outwards. The temperature becomes more evenly distributed. At the entrance, the difference between the wall temperature and the fluid temperature is large for the reason that the wall temperature and the fluid boundary layer temperature near the wall are relatively low. Along the flow direction, due to the heat exchange between the wall and the fluid, the difference between the wall temperature and the fluid temperature gradually decreases and tends to be uniform. The twisted band inserted in the pipeline causes the fluid to rotate and produce secondary flow, which is motion caused by the superposition of forced vortex and axial flow. Under the action of centrifugal force, the fluid in the center of the pipeline and the boundary layer fluid on the wall are fully mixed, and the axial velocity field is homogenized, and thus the heat transfer is enhanced.

The temperature distribution under the same Y but different Re at the pipe cross section is shown in Figure 13. When the Re is larger, the residence time of the particles at a certain position is shorter, which leads to the untimely heat transfer between the fluids. As a result, the particles are concentrated in the two vortex centers, while the swirl flow intensity attenuation is slower and the fluid temperature is higher.

The pipeline temperature distribution of different Y at the same distance under Re = 24,000 is shown in Figure 14. It can be seen from the figure that the temperature field distribution diagram changes from a bimodal peak to a unimodal peak with the increase of torsion. The twist rate is smaller, the swirl flow strength is greater, and the particles are more likely to concentrate on the pipe wall, which increases the probability of collisions between the particles and the wall and improves the heat transfer between the fluid and the wall.

Nu is a dimensionless number that represents the intensity of convective heat transfer. It is the ratio of thermal resistance of the bottom layer of fluid laminar flow to thermal resistance of the convective heat transfer. Its calculation formula is below. Therefore, Nu can be used to compare the effects of twist rate, particle concentration, and flow velocity on enhanced convective heat transfer.
(15)$$Nu=\frac{\alpha L}{\lambda }$$
where, *L* is the characteristic length of the heat transfer surface, which is the pipe diameter (D), in meters; α is the convective heat transfer coefficient (W/(m^2^·k)); and λ is thermal conductivity (w/(m·k)).

It can be seen from Figure 15, Figure 16 and Figure 17 that the heat transfer efficiency of hydrate slurry in the pipeline is mainly affected by twist rate, Reynolds number, and volume fraction. In the segment with the twisted band (0~20 D), the Nu increases firstly, while in the segment without the twisted band (20~50 D), the Nu decreases, and the slowing rate decreases continuously. Nu goes down at a slower rate than Nu goes up. The reason for this is that the tangential velocity and the radial velocity are increased in the twisted band section due to the quadratic distribution of the swirl flow. The convective heat transfer efficiency also increases with the increase in the disturbance between the fluids. However, due to the rapid attenuation of the swirl number, the turbulence intensity decreases and the heat transfer efficiency decreases rapidly. Moreover, it can be seen that the decrease in the Nu number tends towards exponential attenuation, which further indicates that the heat transfer efficiency of hydrate slurry is closely related to the attenuation of swirl flow.

The distribution of Nu on the wall with different twist rates under the same Re and volume fraction is shown in Figure 15. It can be seen from the figure that the initial Nu of the three twist rates is basically the same in the initial area of the front segment (5D), but the Nu is still higher than that outside the twisted band. The Nu gap keeps increasing as the distance increases. The reason for this is that the swirl flow is not stable in the initial region, so the initial swirl number difference is small. However, with the increase in distance, the twist rate is smaller, the centrifugal force is stronger, the forced vortex is stronger, and the probability of particles being thrown towards the wall is increased, so the heat exchange capacity between the wall and the fluid is stronger. On the one hand, the fluid and wall temperature in the rear section tends to be uniform; on the other hand, the attenuation of swirl flow makes the heat transfer efficiency decrease continuously. By comparing the changes in Nu number with the different twist rates, it is found that when the twist rate is smaller, the Nu is larger. However, compared with the heat transfer efficiency from 8.8 to 7.4, the heat transfer efficiency from 7.4 to 6.2 decreases. At the same time, the Nu can be improved by 50~100% compared with the ordinary flat transmission heat efficiency outside the twisted band. This shows that the swirl flow increases the heat transfer efficiency between fluids.

The distribution of Nu on the wall of different Re at the same twist rate and volume fraction is shown in Figure 16. It can be seen from the figure that the Nu increases with the increase in Re, and also the Nu in the back section of the pipe gradually flattens out with the attenuation of the swirl flow. When Re = 12,000, the Nu is 20% higher than that when Re = 6000. When Re = 24,000, the Nu is 30% higher than when Re = 6000. When the Re is greater, the turbulence intensity is greater, and the heat transfer efficiency with the wall surface is stronger. Since the twisted band makes the fluid produce a swirl flow in the pipe and the decay of the swirl number is influenced by Re, the increase in Re improves the motion of solid phase particle entrainment. Therefore, the volume concentration of solid phase particles increases near the pipe wall. The collision frequency is higher, which improves the heat transfer between the fluid and the wall.

The distribution of Nu number on the wall with different volume fractions under the same twist rate and Re is shown in Figure 17. It can be seen from the figure that the average heat transfer performance of the pipeline is enhanced with the increase in the volume fraction of hydrate. However, the increase in wall Nu gradually slows down with the increase in particle volume fraction. This reason is that the increase in particle concentration increases the probability of particle collision with the wall surface, so the heat transfer efficiency increases. Although hydrates are not easy to aggregate by swirl flow, the particles are mainly distributed uniformly around the pipeline wall by centrifugal force. The relative heat transfer area decreases with the increase in concentration. In addition, the concentration is greater, and the inhibition of turbulence will be stronger, so the increase rate of heat transfer efficiency will slow down.

### 3.4. Vortex Line Distribution

Swirl flow is a special kind of turbulent flow that is produced through the superposition of the vortex line and streamline. Vorticity is defined by the angular velocity of a point fluid particle in the flow field, and it represents the curl of each point in the flow field. The vector expression of vorticity is as follows:(16)$$w_{x}\vec{i}+w_{y}\vec{j}+w_{z}\vec{k}=\left(\frac{\partial u_{z}}{\partial y}-\frac{\partial u_{y}}{\partial z}\right)\vec{i}+\left(\frac{\partial u_{x}}{\partial z}-\frac{\partial u_{z}}{\partial x}\right)\vec{j}+\left(\frac{\partial u_{y}}{\partial x}-\frac{\partial u_{x}}{\partial y}\right)\vec{k}$$
where Ω is vorticity:(17)$$\Omega =2w=\sqrt{w_{x}^{2}+w_{y}^{2}+w_{z}^{2}}$$
where w is the rotational angular velocity of the fluid and i, j, and k are the unit vector represented at the three coordinate axes.

According to the definition of a vortex line, it can be seen that the vortex line is tangent to the vortex vector everywhere. Each fluid micromass rotates tangentially around the vortex line. The vortex lines at different positions under the same Y and Re are shown in Figure 18. It can be seen from the figure that the rotation direction of the particle is the same as that of the twisted band. At first, the hydrate particles begin to swirl near the two ends of the twisted band, and two symmetrical vortex centers appear. The center of the vortex begins to shift toward the center near the twisted band, and then shifts laterally toward the side wall of the pipe. The swirl flow intensity begins to attenuate in the pipe outside the twisted band and the two vortex centers gradually begin to merge. The vortex center and velocity center do not coincide at first, and the velocity center and vortex line center basically coincide when the vortex is quite stable. The swirl flow causes hydrate particle suspension outside and makes the hydrate particle itself swirl. The particles of liquid bridge force are not easy to gather by swirl flow, and particle concentration distribution is more even. Thus, it is not easy to form a reunion and deposition in the pipeline.

The vortex diagram of different Re at the same Y is shown in Figure 19. Re is one of the main factors affecting the attenuation rate of swirl flow. It can be seen from the figure that the Re is larger, and the merging distance between the two symmetric vortex line centers is farther, which indicates that the vortex strength attenuation is slower. The velocity center merges at the same location as the vortex line center merges.

The vortex diagram at the same position (Re) but different Y is shown in Figure 20. It can be seen from the figure that the twist rate is smaller, the swirl strength is greater, and flow is more stabilized. However, compared with Figure 19, the position at the center of the vortex line where the merger occurs is the same, and it indicates that the twist rate has no obvious influence on the attenuation rate of swirl strength. Therefore, the attenuation of swirl flow strength is mainly affected by Re. While the twist rate has a great influence on the initial swirl strength, it has little influence on the attenuation section of swirl flow outside the twisted band.

### 3.5. Attenuation Law of Wall Shear Stress

The hydrate particles and pipelines generate shear stress under the action of swirl flow, and shear stress is an important factor to ensure that the hydrate particles are not easy to deposit and “suspend” in the pipeline. As shown in Figure 21, it is found that as the Y gets smaller, the shear stress becomes greater under the same Re. As shown in Figure 22, it is found that as the Re becomes greater, the shear stress becomes greater under the same Y. Comparing Figure 21 and Figure 22, it is found that both Y and Re have an impact on the shear stress, but Re has a larger impact. The reason is that the twist rate mainly affects the initial shear stress. Although the initial shear stress has an effect on the posterior segment over a large distance, its attenuation rate is almost the same. The Re not only affects the initial shear stress, but also affects the attenuation rate of the entire attenuation segment. In addition, it can be observed that the attenuation of shear stress in the initial stage is similar to the attenuation of the swirl number, which shows an exponential attenuation. Najafi [8] concluded that the shear force of the fluid and the wall is the reason for the reduction in the swirl number. However, when the swirl flow attenuates to a certain extent, it can be observed that the attenuation of shear stress at the end of the pipe with weak swirl flow in Figure 21 and Figure 22 is no longer exponential but tends to be first-order function attenuation. The reason is that all hydrates are suspended in the pipe with strong swirl flow, and the shear stress is only related to the attenuation of swirl flow. However, the hydrate particles begin to deposit in the rear section of the pipe with weak swirl strength and the shear stress of the deposited part changes, so it is different from exponential attenuation.

### 3.6. Attenuation Law of Swirl Number

Improving the initial swirl number and reducing the swirl flow attenuation efficiency is the key to improving the hydrate transport efficiency. Therefore, it is of great significance to study the attenuation law of the swirl number. Swirl number (Ω) is one of the important parameters for characterizing the swirl flow strength of hydrate, and its expression is as follows:(18)$$\Omega =\frac{\mathrm{Tangential}\mathrm{momentum}}{\mathrm{Axial}\;\mathrm{momentum}}=\frac{\int_{0}^{R}\rho u_{z}u_{\theta }r^{2}dr}{R\int_{0}^{R}\rho u_{z}^{2}rdr}$$

Suppose the angular velocity c is the same at every point on the same section:(19)$$u_{\theta }=cr\;\;\;\;\;\;\frac{u_{\theta }}{u_{m\theta }}=\frac{r}{R}$$

The expression of the swirl number is:(20)$$\Omega =\frac{\int_{0}^{R}\rho u_{z}u_{m\theta }r^{3}dr}{R^{2}\int_{0}^{R}\rho u_{z}^{2}rdr}=\frac{u_{m\theta }}{2u_{z}}$$
where, R is the pipe radius, ρ is the fluid density, $u_{\theta }$ is the tangential velocity, $u_{Z}$ is the axial velocity, c is the rotational angular velocity, and $u_{m\theta }$ is the maximum tangential velocity.

It can be seen from Figure 23 and Figure 24 that the attenuation of swirl number decreases in an exponential manner. It can be seen from Figure 23 that the decay rate decreases with the increase in Re under the same Y. At the same time, the swirl number at the initial position is almost the same, which indicates that the change of Re has almost no influence on the initial swirl number. It can be seen from Figure 24 that Y is the smaller, and the attenuation of the swirl number is slower under the same Re. At the same time, the swirl number at the initial position is different, the twist rate Y is smaller, and the initial swirl number is larger. Therefore, the initial swirl number is mainly affected by Y, while Re has almost no influence on the initial swirl number. After passing through the swirler, the swirl flow attenuates rapidly. Although the twist rate Y affects the swirl number at the same position, its attenuation rate is almost unchanged. Its attenuation rate is mainly related to Re, while Y has little influence on swirl number.

KreithSonju, Wolfeta, and Baker Sayre [28] et al. conducted an experimental study on the attenuation of swirl flow, and Najafi [8] et al. conducted a theoretical derivation on the attenuation of swirl flow, and the results all showed that the attenuation of swirl flow number in straight pipe flow is basically exponential. Its expression is:(21)$$\frac{\Omega }{\Omega_{0}}=e^{-\beta \left(\frac{L}{D}\right)}$$
(22)$$Ln\left(\frac{\Omega }{\Omega_{0}}\right)=-\beta \left(\frac{L}{D}\right)$$
where Ω is the swirl number at different positions; $\Omega_{0}$ is the initial swirl number; L is the distance from the initial position; D is the pipe diameter; and β is the intensity attenuation index of swirl number.

The dimensionless swirl number is obtained as shown in Figure 25 and Figure 26. Figure 25 shows the dimensionless swirl number in the different position under different Y. It can be seen from the figure that the decay velocity of swirl flow under different twist rates is almost the same. Therefore, the twist rate only affects the initial swirl number, and it has almost no effect on the swirl flow attenuation. Figure 26 shows the dimensionless swirl number in the different position under the same Y and different Re. It can be seen from the figure that Re is a major factor affecting the attenuation of swirl flow, the Re is smaller, and the influence of Re on the attenuation of swirl flow is stronger. The swirl flow attenuation exponential under different Re conditions is calculated, and the relation between these coefficients and Reynolds number is shown in Figure 27.

The fitting curve can be obtained by fitting the logarithmic data points, the deviation of which is less than 10%. The relationship between Re and β is as follows:(23)β=0.33968−0.02974ln(Re)

### 3.7. Deposition Law of Hydrate Particles

The concentration distribution diagram of hydrate particles is shown in Figure 28 by short twist band with a twist rate of 8.8 and the axial velocity at the inlet is 3 m/s. Compared with normal direct flow, swirl flow has a stronger carrying capacity for solid particles, and it has the characteristics of uniform particle concentration distribution, low energy consumption, and a long distance from transport. Swirl flow velocity can be considered the combination of tangential velocity, axial velocity, and radial velocity, that is the combination of forced eddy current and flat direct current. Tangential velocity rotates hydrate particles, and tangential force makes solid particles “swirl” in the pipeline. The particles move in concentric circles, and the axial velocity provides the impetus for the forward transport of the particles. The combination of the two can improve the transportation distance of hydrate particles in the pipeline. The hydrate particles are distributed near the pipe wall at the beginning under the centrifugal force generated by swirl flow. Subsequently, the swirl flow strength is continuously attenuated, centrifugal force is less than gravity, and particles cannot be uniformly distributed around the pipeline wall so they begin to sink. The radius of swirl forward orbit of particles decreases, so the concentration at the center of the pipe begins to increase partly. The concentration distribution diagram of particles at the 100D position is shown in Figure 29. With the further attenuation of swirl strength, particles begin to sink and gather.

The particle concentration distribution in the same cross section under different twist rates is as shown in Figure 30. The particles are mainly concentrated at the bottom of the pipe cross section outside the twisted band, and the particle concentration in the upper part is almost zero. The accumulation of hydrate particle concentration will increase with time, and the possibility of blockage greatly increases, which affects the safe operation of the pipeline. It can be seen that the twist rate is smaller, and the circumferential distribution of the particles is more uniform. When the twist rate is 8.8, the particle concentration at the lower part of the pipeline was higher than that at the upper part, and partial deposition of particles occurred. However, the concentration in the cross section of normal direct current transport is relatively uniform. When the twist rate is 7.4, the carrying efficiency is obviously greatly improved. The carrying efficiency improves but the improvement amplitude is slowed down with the further decrease in the twist rate. The particle carrying efficiency is twice as high. This shows that the swirl flow can prevent hydration aggregation and expand the safe transport boundary.

Another important factor affecting particle deposition is fluid flow velocity. The flow velocity is greater, the turbulence intensity is greater, the pulsation is more violent, and the particles are more difficult to deposit. The distribution diagram of particle concentration in the vertical radial direction of the same section under the same twist rate and different Re is shown in Figure 31. It can be seen that the hydrate particles are deposited in the form of a “U”, and that the concentration of particles is higher at the side wall of the pipeline and lower in the center. The reason is that the Re is larger, the swirl flow attenuation is slower, and the tangential velocity is greater. Under the action of centrifugal force, particles are more likely to be thrown to the side wall. In addition, the Re is larger, the cross-section particle distribution is more uniform, and the particle concentration is smaller. However, the Re is higher and the higher efficiency of hydrate carrying efficiency is reduced. Therefore, an appropriate twist rate and flow velocity can not only improve the hydrate carrying efficiency, but also save energy consumption.

## 4. Conclusions

DPM and RSM models are used to study the transportation of hydrate in pipelines under the condition of swirl flow attenuation with a short twisted band. The temperature distribution, velocity field, turbulence intensity, swirl flow attenuation law, and movement distribution of hydrate particles of slurry are analyzed. The conclusions are as follows:The swirl flow velocity presents a symmetrical bimodal structure. The two velocity centers gradually move closer to the center of the pipeline and finally merge together with the attenuation of the swirl flow. The axial velocity is an “M” shape in the twisted segment, the peak value is 1/2r away from the pipeline wall, and the axial velocity is a parabolic shape in the rear pipeline segment. The absolute value of radial velocity is relatively small, which is the result of the redistribution of velocity by twisted band and it rapidly drops to 0 m/s in the posterior segment. The tangential velocity is the “M” shape in the twisted band section and the back pipe section. The peak value appears 1/6~1/7r away from the pipe wall.Swirl flow can improve the heat transfer efficiency between the pipeline wall and the fluid, which is mainly related to Reynolds number, twist rate, and particle concentration. In the twisted band section, the Nu increases firstly, and it is removed after the twisted band. The Nu decreases and the slowing rate decreases continuously. The increase in Re has a more obvious induction effect on the motion of solid particles, thus Nu increases. With the increase in the volume fraction of particles, the increase rate of Nu number on the wall slows down. The twist rate is smaller, the Nu is larger, and the heat efficiency is higher.The turbulence intensity distribution in the twisted band section shows a “W” shape at the center of the rigid main vortex. The vortex in the free vortex area near the twisted band has a small scale and large unstable velocity pulsation, so the turbulence intensity is large. In the merging process of two symmetric vortexes outside the twisted band, the pulsation velocity at the central vortexes is increased, and the turbulence intensity distribution curve shows a “U” shape.The swirl direction of hydrate particles is the same as that of the twisted band. The vortex line swirl center begins to appear at both ends of the proximal twisted band, then it moves to the center of the proximal twisted band, and it finally moves to the edge of the pipeline to achieve stability. After leaving the twisted band, the vortex attenuates rapidly. The twist rate Y is smaller, the Re is larger, and the vortex attenuates more slowly. The attenuation rate of vorticity is mainly affected by Re, while the twist rate mainly affects the initial vorticity size. The shear stress is the main reason for the decrease in swirl strength, and the shear stress decreases exponentially in the pipe section with strong swirl flow. The twist rate mainly affects the initial swirl number, but it has little influence on the attenuation rate of swirl flow. The twist rate is smaller, the initial swirl number is larger. The attenuation of swirl flow is mainly related to Re, and the Re is larger, and it is slower. The swirl flow decreases exponentially, and the relation expression between swirl flow attenuation exponent and Re is obtained.The hydrate particles are distributed near the pipe wall under the action of centrifugal force generated by the swirl flow. The hydrate particles do not enter the forced vortex region. Due to the effect of shear force, the carrying distance of particles is increased. However, the swirl flow rapidly attenuates with the increase in carrying distance, the swirl radius of the particles decreases, and deposition occurs at the end of the pipe. The twist rate is larger, the swirl flow intensity is smaller, the attenuation is faster, and the particles are more likely to accumulate. In addition, the Re is larger, the cross-section particle distribution is more uniform, and the particle concentration is smaller.

## Figures and Tables

**Figure 1 entropy-23-00913-f001:**
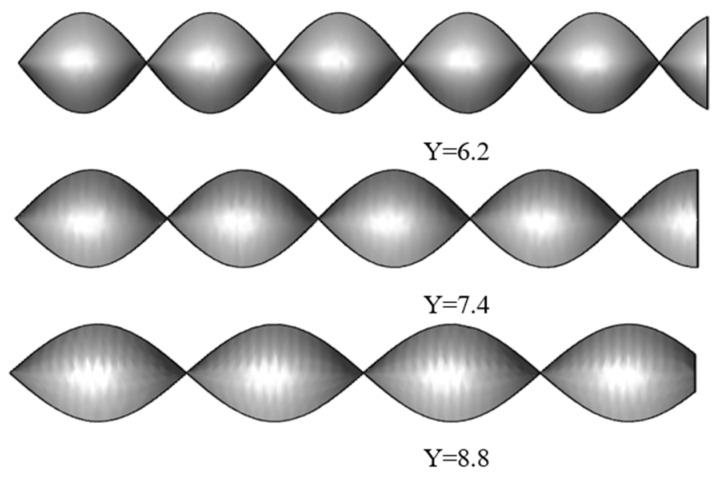
Twisted band model.

**Figure 2 entropy-23-00913-f002:**
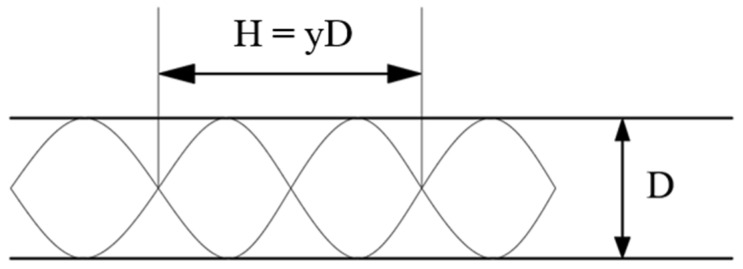
Twist rate schematic diagram of twisted band.

**Figure 3 entropy-23-00913-f003:**
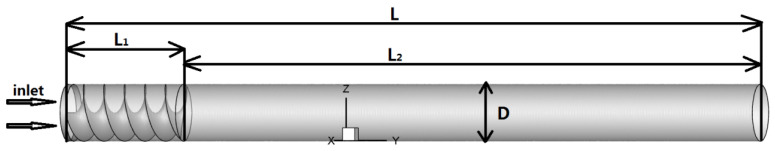
Physical model.

**Figure 4 entropy-23-00913-f004:**
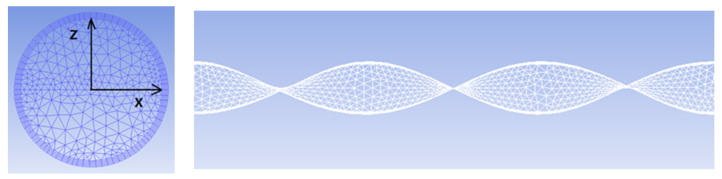
Grid division diagram of the front pipe segment.

**Figure 5 entropy-23-00913-f005:**
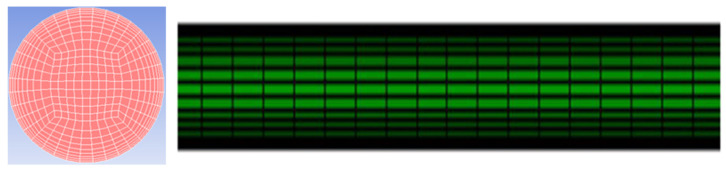
Grid division diagram of the rear pipe segment.

**Figure 6 entropy-23-00913-f006:**
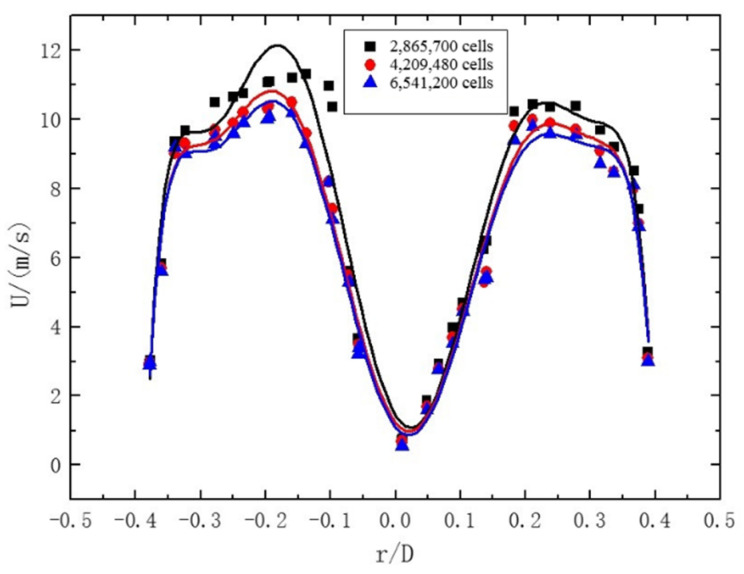
The velocity distribution at pipeline section Z = 5D with different mesh sizes.

**Figure 7 entropy-23-00913-f007:**
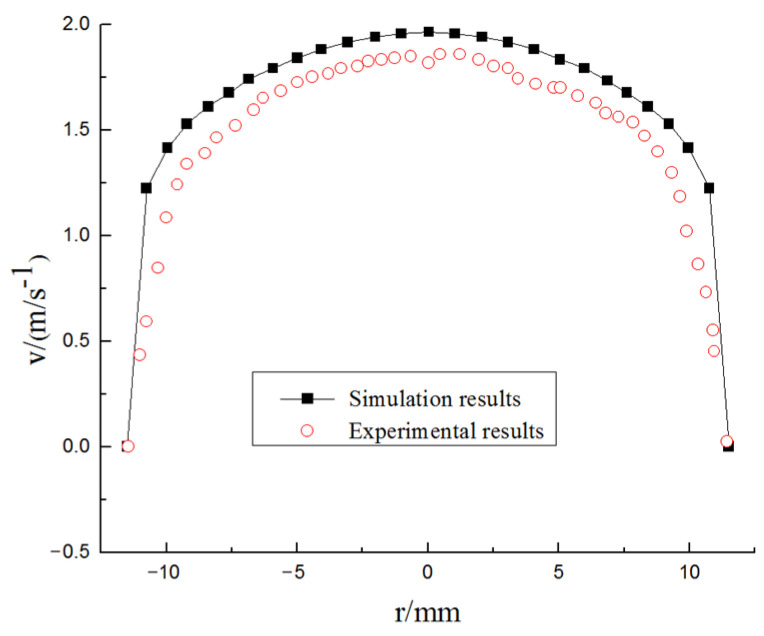
Velocity field distribution.

**Figure 8 entropy-23-00913-f008:**
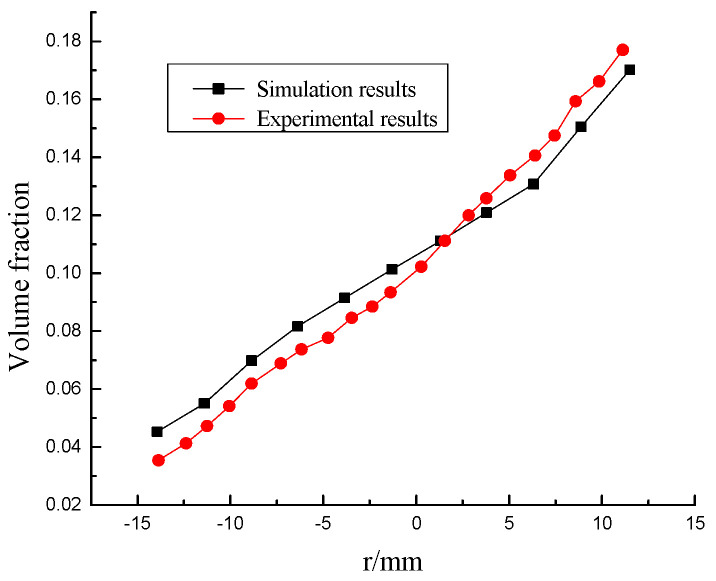
Volume fraction distribution.

**Figure 9 entropy-23-00913-f009:**
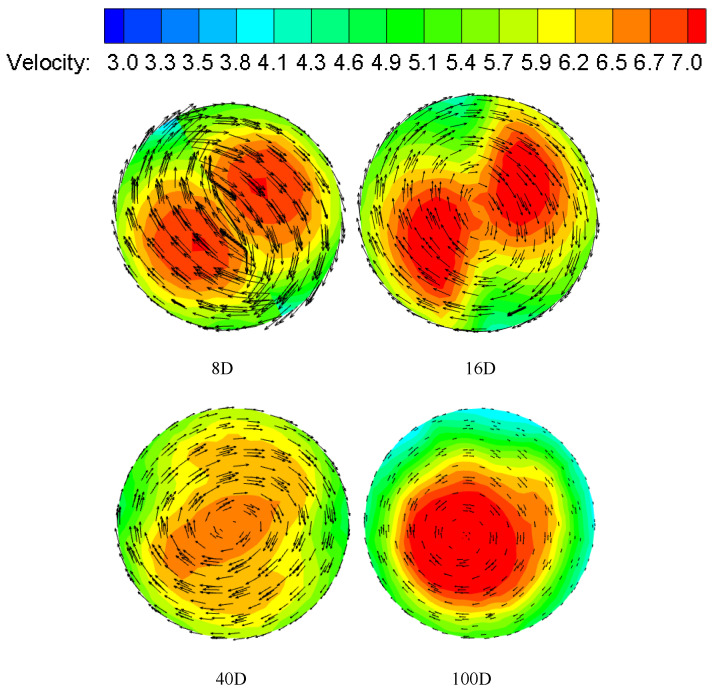
Velocity and velocity vector distributions at different sections. Re = 24,000, Y = 6.2.

**Figure 10 entropy-23-00913-f010:**
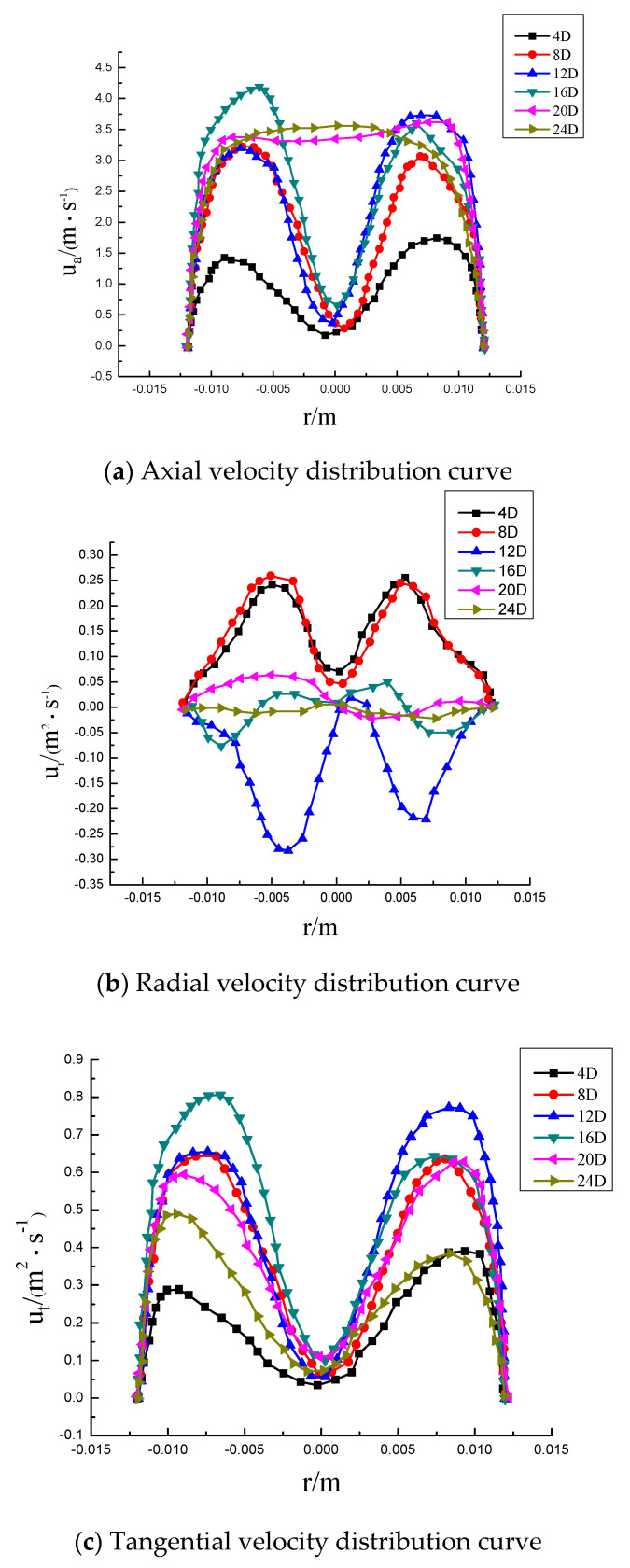
Velocity distribution at each position on the center line of different sections. Re = 6000, Y = 6.2.

**Figure 11 entropy-23-00913-f011:**
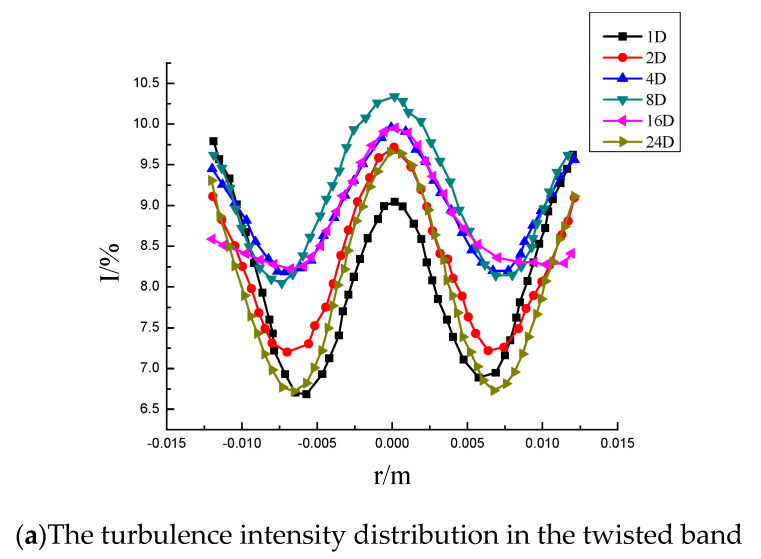
Distribution of turbulence intensity at different axial sections of the pipeline.

**Figure 12 entropy-23-00913-f012:**
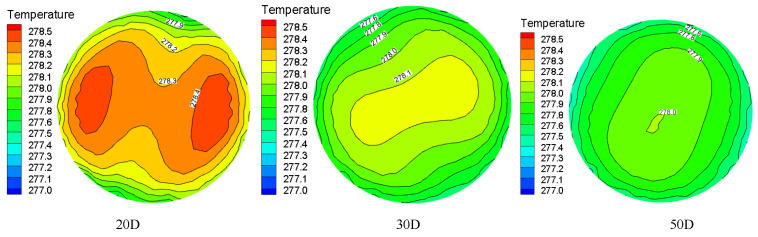
Temperature distribution at different pipe sections under the same Re and Y.

**Figure 13 entropy-23-00913-f013:**
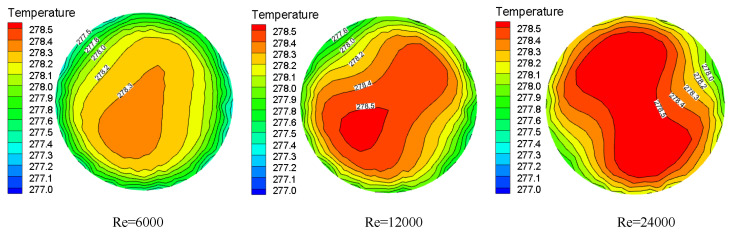
Temperature distribution under the same torsion rate and pipe section but different Re.

**Figure 14 entropy-23-00913-f014:**
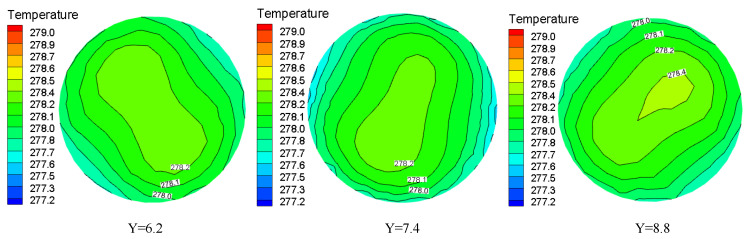
Temperature distribution under the same Re and pipe section but different Y.

**Figure 15 entropy-23-00913-f015:**
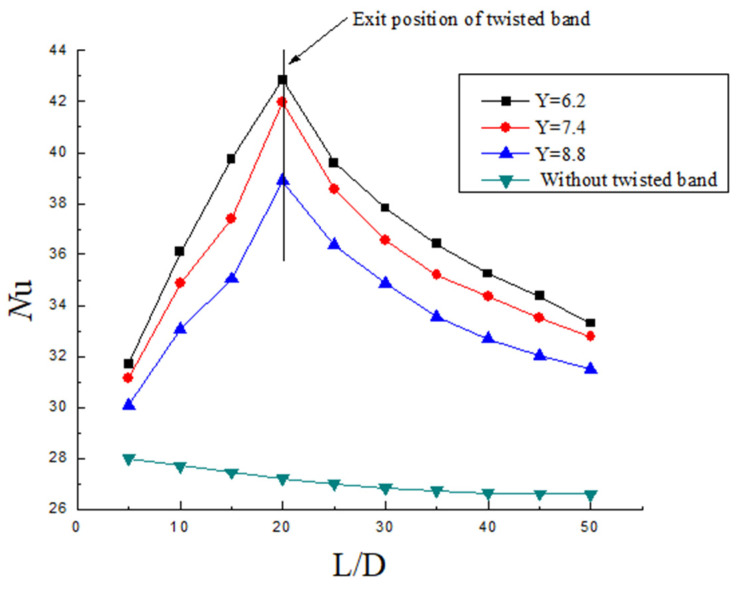
Nusselt number distribution under the same Re and volume fraction but different Y.

**Figure 16 entropy-23-00913-f016:**
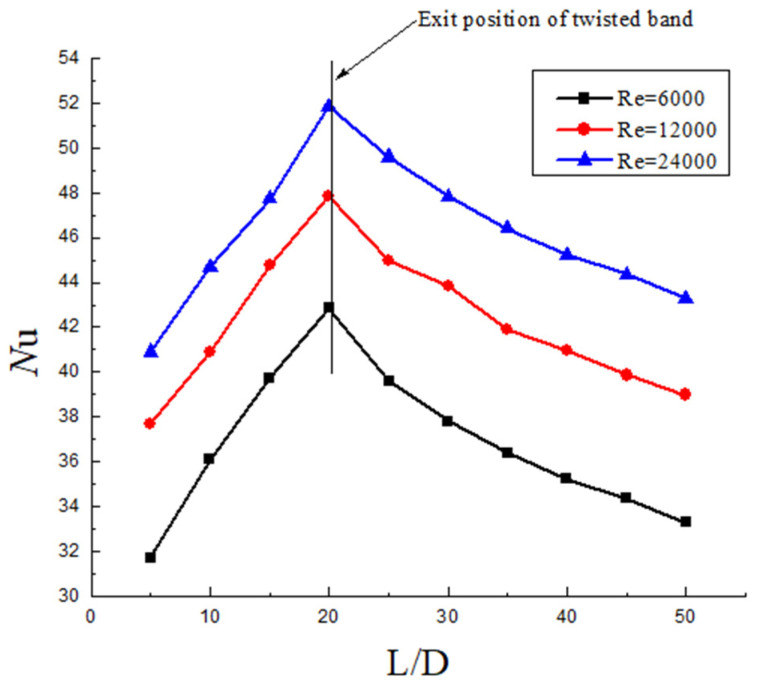
Nusselt number distribution under the same Y and volume fraction but different Re.

**Figure 17 entropy-23-00913-f017:**
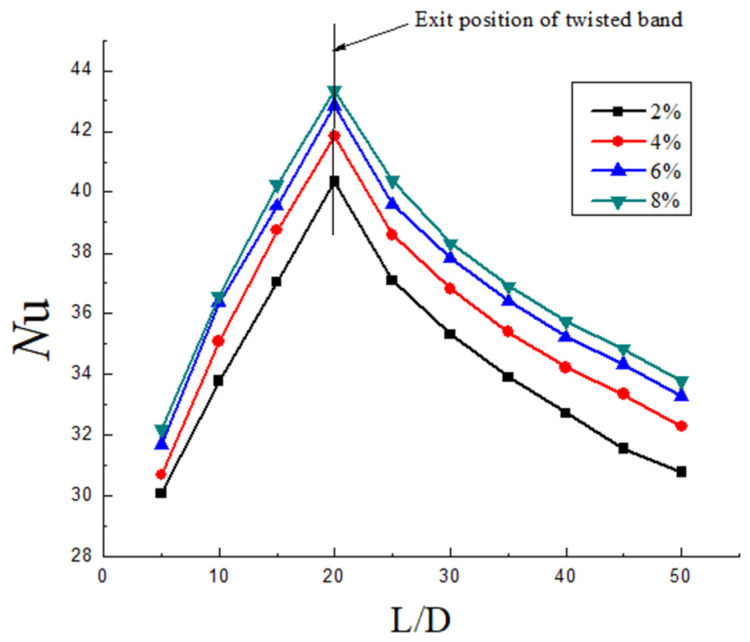
Nusselt number distribution under the same Y and Re but different volume fraction.

**Figure 18 entropy-23-00913-f018:**
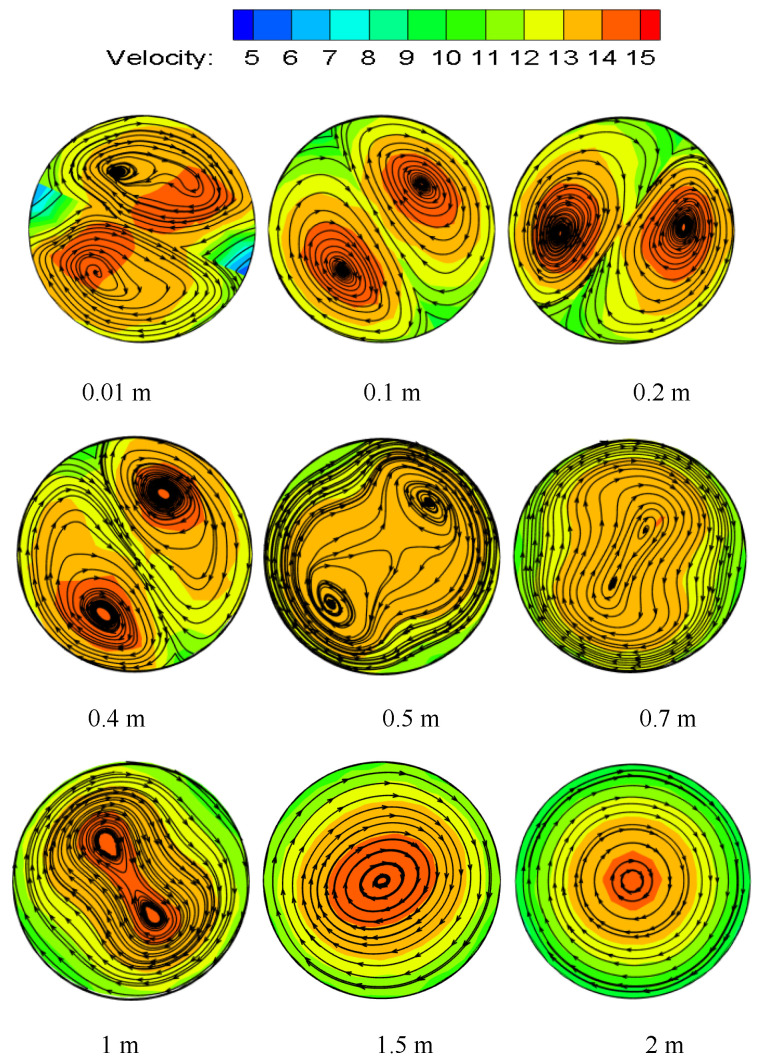
Vortex lines at different positions under the same Y and Re.

**Figure 19 entropy-23-00913-f019:**
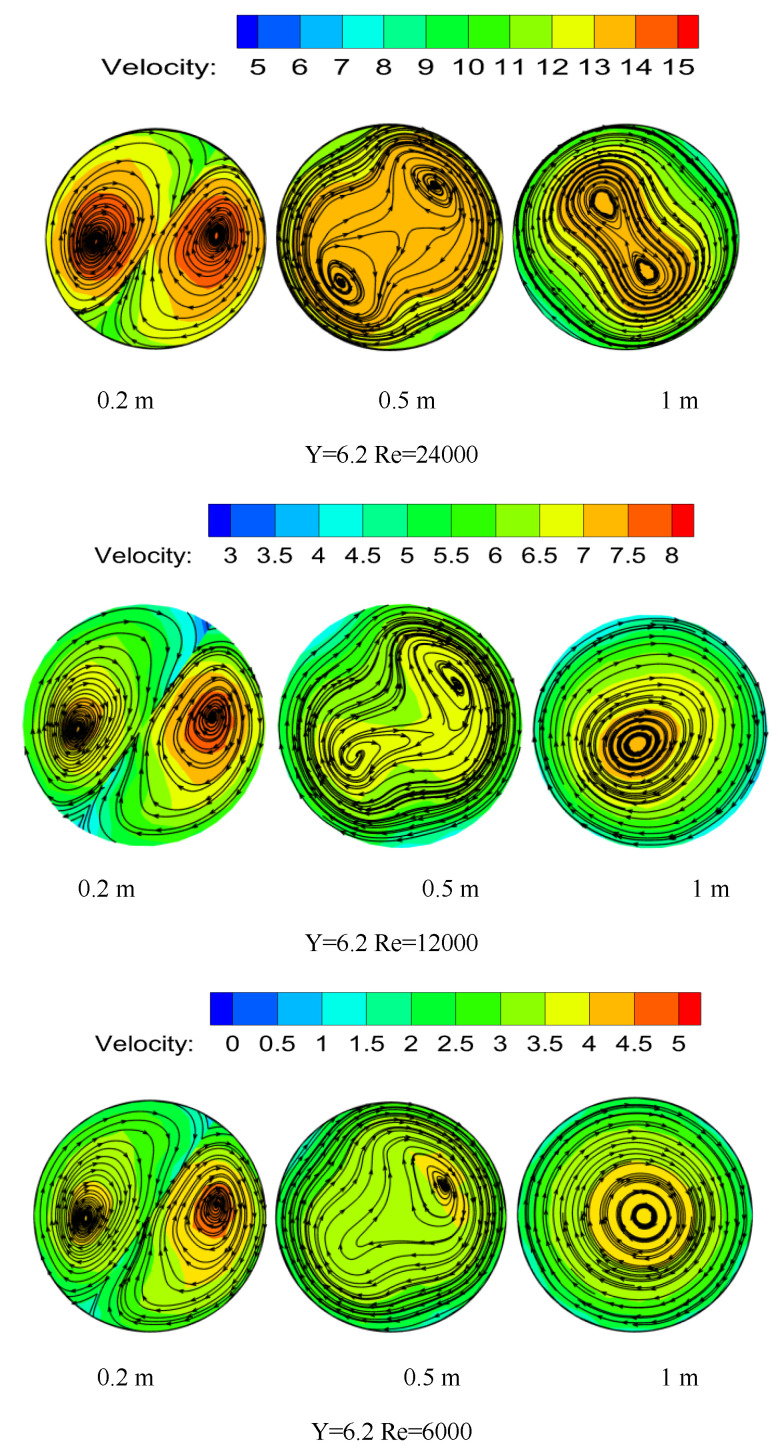
Vortex lines under the same position and Y but different Re.

**Figure 20 entropy-23-00913-f020:**
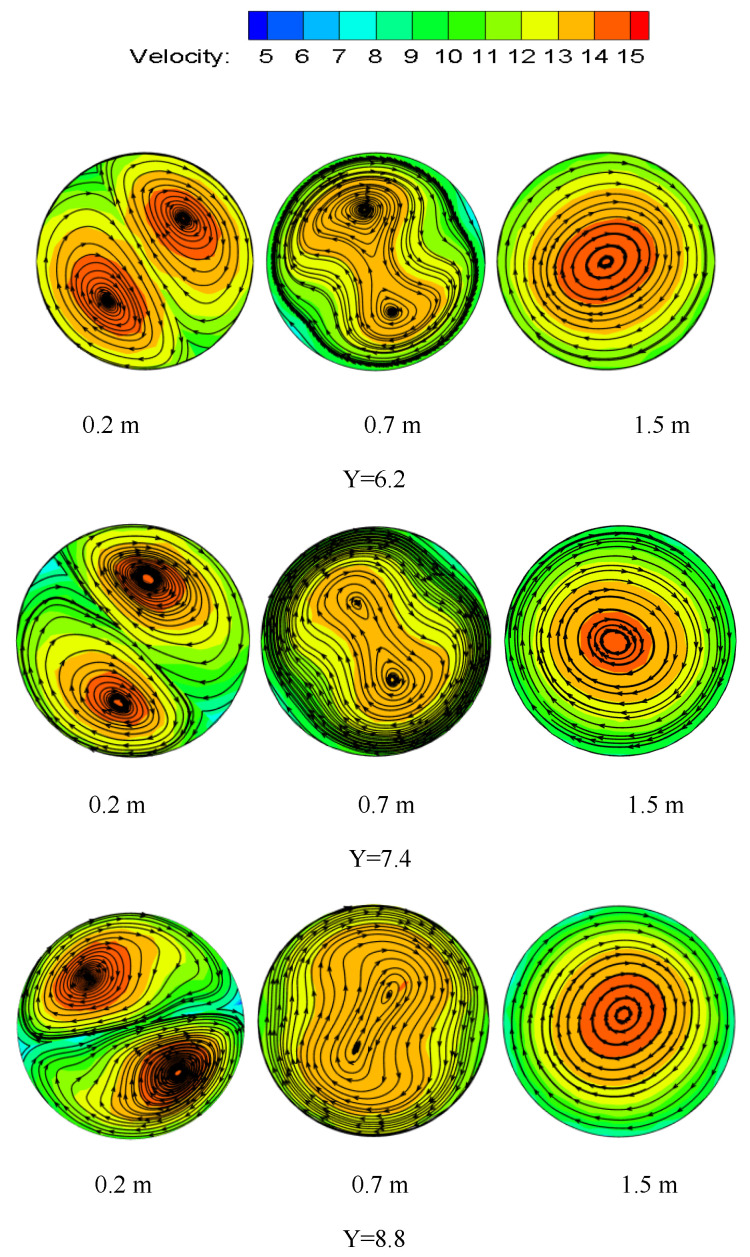
Vortex lines under the same position and Re but different Y.

**Figure 21 entropy-23-00913-f021:**
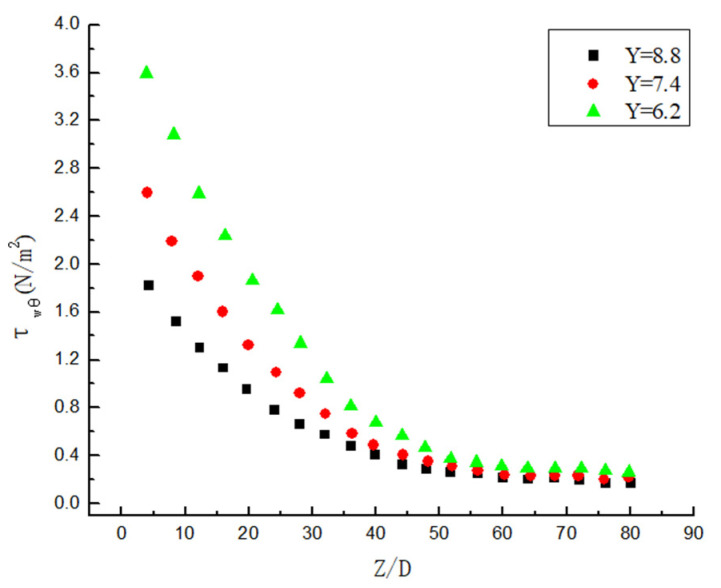
The shear stress change under the same Re but different Y.

**Figure 22 entropy-23-00913-f022:**
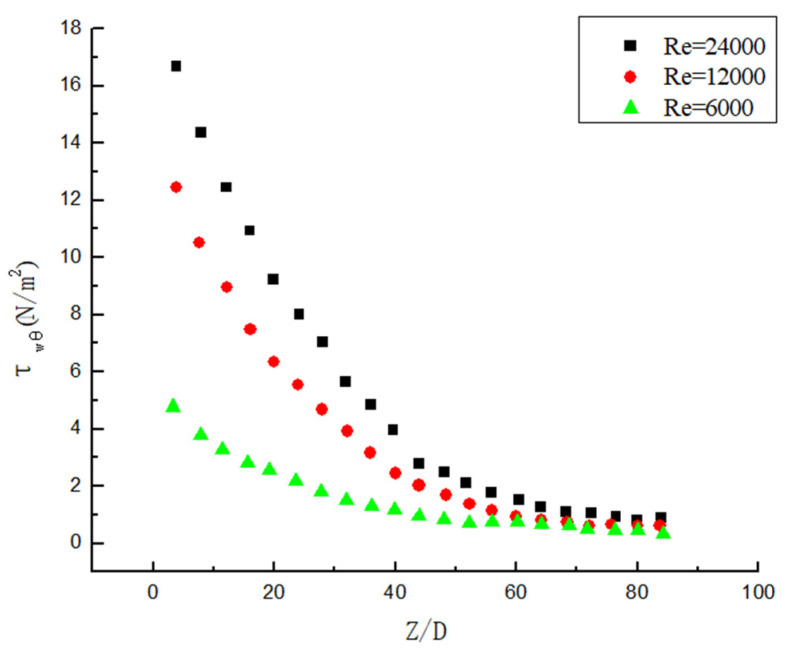
The shear stress change under the same Y but different Re.

**Figure 23 entropy-23-00913-f023:**
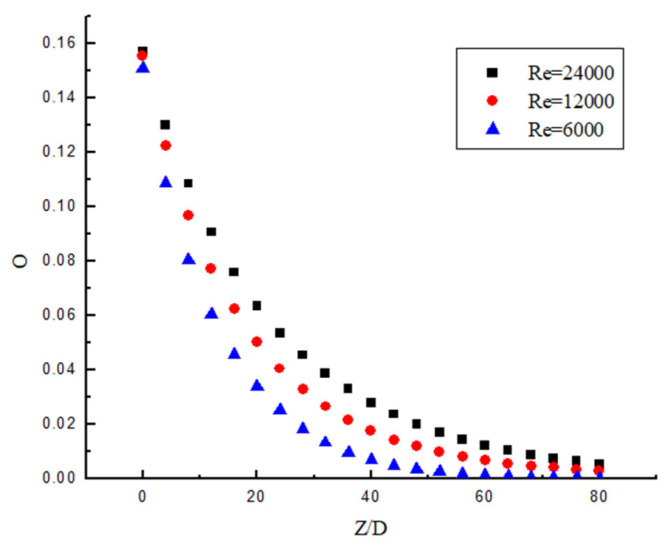
The swirl number change under the same Y but different Re.

**Figure 24 entropy-23-00913-f024:**
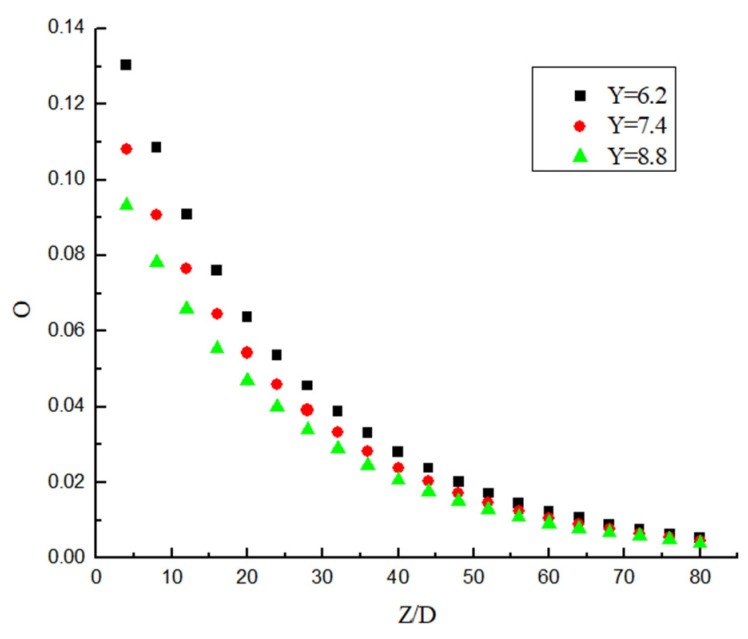
The swirl number change under the same Re but different Y.

**Figure 25 entropy-23-00913-f025:**
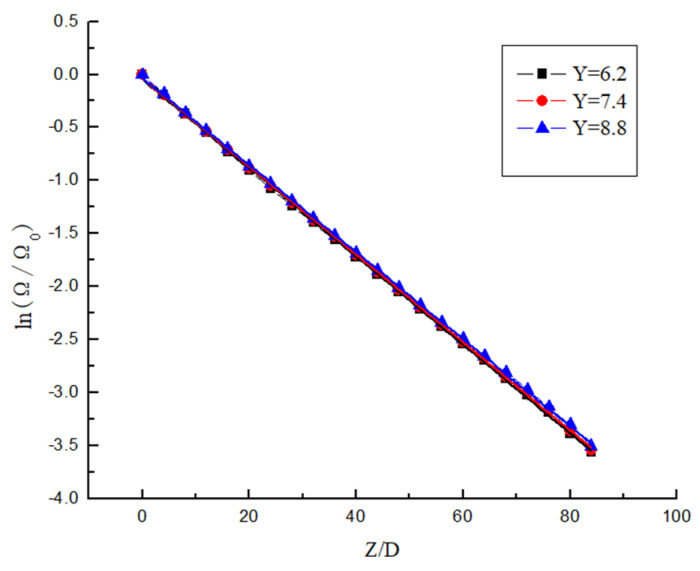
The dimensionless swirl number variation under different Y.

**Figure 26 entropy-23-00913-f026:**
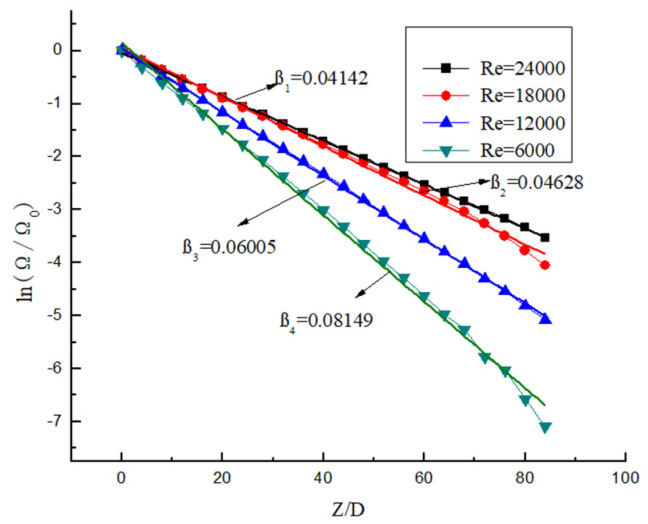
The dimensionless swirl number changes under different Re.

**Figure 27 entropy-23-00913-f027:**
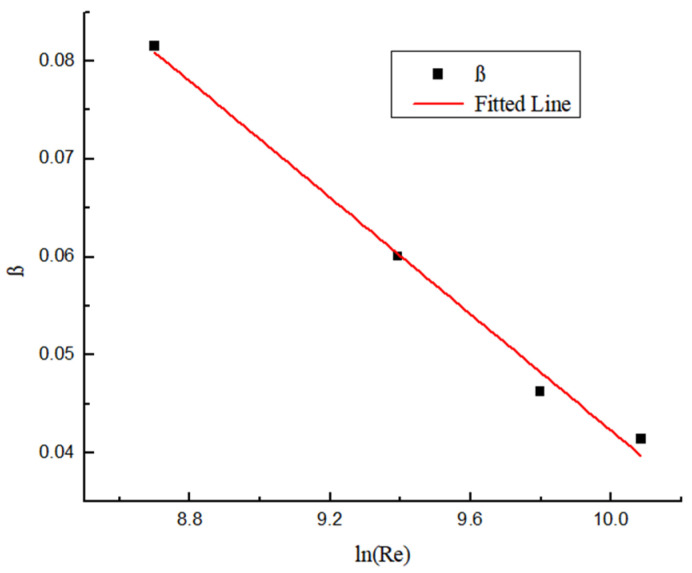
The relationship between Re and β.

**Figure 28 entropy-23-00913-f028:**
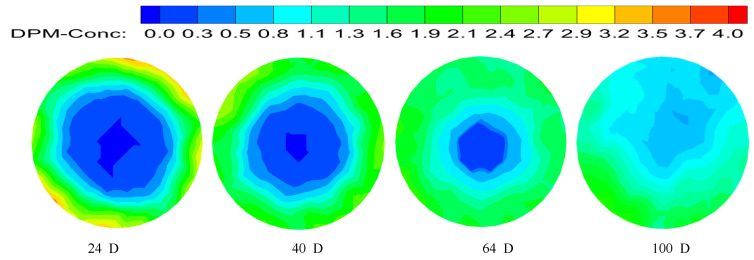
Hydrate particle concentration distribution at different location.

**Figure 29 entropy-23-00913-f029:**
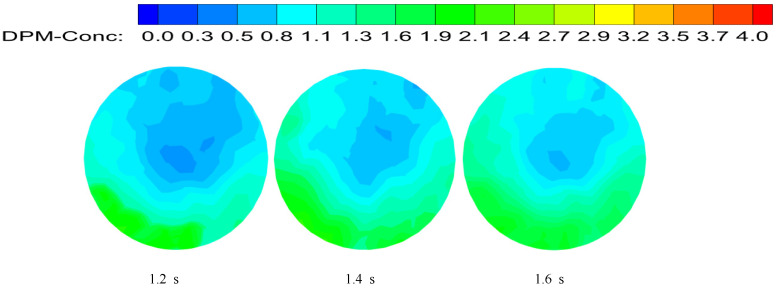
Particle concentration distribution at different time.

**Figure 30 entropy-23-00913-f030:**
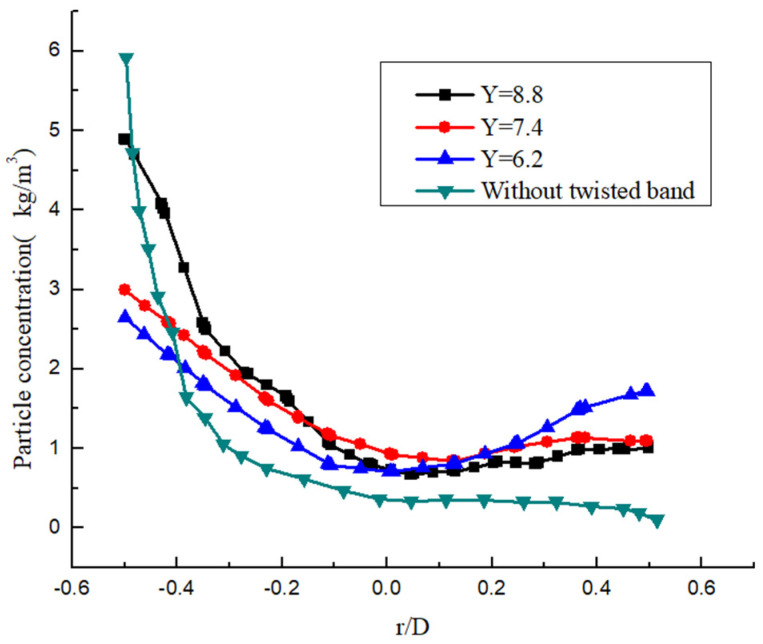
Particle concentration distribution under different Y.

**Figure 31 entropy-23-00913-f031:**
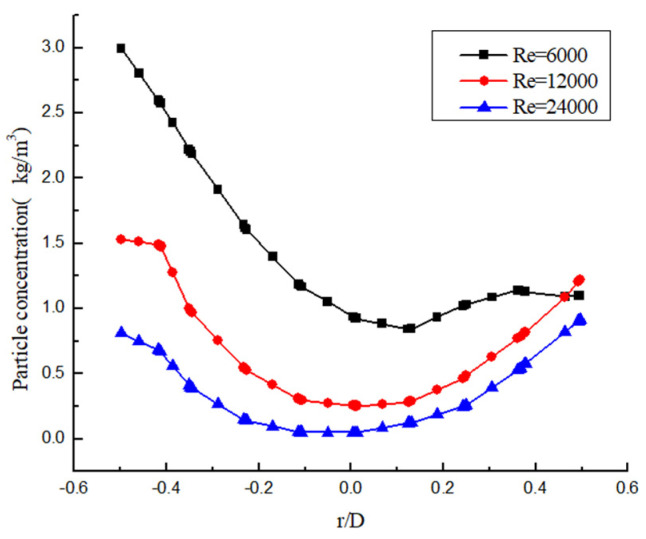
Particle concentration distribution under different Re.

**Table 1 entropy-23-00913-t001:** Parameters of numerical simulation.

Particle Concentration (%)	Particle Size (mm)	Twist rate Y	Initial Velocity (m/s)
1~8	0.001	6.2/7.4/8.8	0.5~12

## Data Availability

The data used to support the findings of this study are available from the corresponding author upon request.

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
