# Peer review of "Numerical Simulation of Swirl Flow Characteristics of CO2 Hydrate Slurry by Short Twisted Band"

_entropy, 2021, doi:10.3390/e23070913_

Round 1
Reviewer 1 Report
Review report on paper 1276568: “Numerical simulation on swirl flow charcateristics of CO2 hydrate by short twisted band”
This paper deals with numerical simulations of swirl flow in a truncated straight pipe with applications to slurry flow transportation. The problem is a relevant one for the oil industry where transport of oil from the production wells to the refining plants is performed by means of horizontal pipeline systems. One important feature of the simulations is the inclusion of heat transfer effects. Although the simulations were performed using a commercial code, the results are valuable because they serve as a comparison with future simulations on slurry flow. Therefore, I think that the paper may be published in Entropy once the authors respond to the comments below.
1) My first comment concerns the language. I urge to improve the English throughout the text.
2) This kind of simulations are relevant for applications to slurry transport in the oil industry. Therefore, it would be nice if the authors provide more details about the inlet boundary conditions. How are they implementing the flow swirling at the pipe entrance? Such details are important for CFD practitioners interested in swirling flow.
3) I have a question about Eq. (2). As written below tau _ij is the viscous stress tensor. Why is the divergence of 1/tau _ij needed in the last term on the right-hand side? What does it mean?
4) I think a convergence test verification is mandatory for this kind of simulations. For example, Figures 5 and 6 shows results for such a test. I wonder how many cells were used? Also, in Figure 5 the error when comparing with the experimental velocity profiles is 20%. I urge the authors to specify the number of cells and to perform a sequence of calculations with improved spatial resolution to investigate the asymptotic approach to convergence.
5) The results in Section 3 are quite interesting and they deserve a better presentation of a grid-independence test as suggested in point 4) above. This will certainly help future CFD applicants for similar simulations to decide on the more suitable parameters.
Author Response
Response to Reviewer 1 Comments
Point 1: My first comment concerns the language. I urge to improve the English throughout the text.
Response 1: I make sentences connected and remove grammar mistakes and typos. The modified contents are in line 12, 13, 28, 29, 78, 144, 203, 212, 228, 231, 232, 233, 238, 265, 281, 287, 317, 346, 356, 360, 379, 516, 553, 617, 626, 635, 641, 654.
Point 2: This kind of simulations are relevant for applications to slurry transport in the oil industry. Therefore, it would be nice if the authors provide more details about the inlet boundary conditions. How are they implementing the flow swirling at the pipe entrance? Such details are important for CFD practitioners interested in swirling flow.
Response 2: The twisted band is used as the swirl device and is set at the entrance of the pipeline, and it is fixed. After the gas-liquid phase passes through the twisted band, the phase interface is reconstructed under centrifugal force. The modified sentence is in line 78.
Point 3: I have a question about Eq. (2). As written below tau _ij is the viscous stress tensor. Why is the divergence of 1/tau _ij needed in the last term on the right-hand side? What does it mean?
Response 3: The reason is the convenience of calculation, and it is better to calculate the result quickly.
Point 4: I think a convergence test verification is mandatory for this kind of simulations. For example, Figures 5 and 6 shows results for such a test. I wonder how many cells were used? Also, in Figure 5 the error when comparing with the experimental velocity profiles is 20%. I urge the authors to specify the number of cells and to perform a sequence of calculations with improved spatial resolution to investigate the asymptotic approach to convergence.
Response 4: The simulation and experimental measurement of ice slurry are relatively perfect, and they have many similarities in many basic properties such as density. Sari et al.'s ice slurry flow experiment and the theoretical value derived by Kitanovski et al. under similar working conditions are adopted as simulation verification.
Point 5: The results in Section 3 are quite interesting and they deserve a better presentation of a grid-independence test as suggested in point 4) above. This will certainly help future CFD applicants for similar simulations to decide on the more suitable parameters.
Response 5: Thank you for your suggestions, and I will use more suitable parameters.

Reviewer 2 Report
The paper under review deals with the research on mixed pipeline of natural gas hydrate plugging problems and to ensure the safety of pipeline operation. The subject is very interesting, it is worthy of investigation. The results in the study are achieved via numerical simulation comprised with experimental data. The article tackles an important issue in chemical engineering and mechanics and also multiphase flows and therefore is suitable for Entropy.
Comments:
The authors rightly noted that the research on two-phase swirl flow is conducted all over the world by experimental tests and numerical simulation. Therefore, in my opinion, the novelty of the research is “hardly visible” (debatable). I think the Authors should indicate the novelty.
The paper is well prepared and written in good English. The article contains adequate and appropriately selected 29 literature items. In opinion of the reviewer the article needs minor correction/complete with data.
Author Response
Response to Reviewer 2 Comments
Point 1: The authors rightly noted that the research on two-phase swirl flow is conducted all over the world by experimental tests and numerical simulation. Therefore, in my opinion, the novelty of the research is “hardly visible” (debatable). I think the Authors should indicate the novelty.
Response 1: Dear reviewers and editors, thank you very much for your valuable suggestions and comments. The current simulation of the flow characteristics of hydrate slurry mainly focuses on the study of ordinary straight pipeline. The heat transfer, pressure drop and deposition law of hydrate slurry in swirl flow system are little, and it is necessary to establish an appropriate model to guide the placement of the rotor and the safe flow range.
Point 2: The paper is well prepared and written in good English. The article contains adequate and appropriately selected 29 literature items. In opinion of the reviewer the article needs minor correction/complete with data.
Response 2: Dear reviewers and editors, thank you very much for your valuable suggestions and comments. I added minor correction about introduction, results and discussions.

This manuscript is a resubmission of an earlier submission. The following is a list of the peer review reports and author responses from that submission.